# ANTIFRAGILE AND ROBUST HETEROSCEDASTIC BAYESIAN OPTIMISATION

## ABSTRACT

Bayesian Optimisation is an important decision-making tool for high-stakes applications in drug discovery and materials design. An oft-overlooked modelling consideration however is the representation of input-dependent or heteroscedastic aleatoric uncertainty. The cost of misrepresenting this uncertainty as being homoscedastic could be high in drug discovery applications where neglecting heteroscedasticity in high throughput virtual screening could lead to a failed drug discovery program. In this paper, we propose a heteroscedastic Bayesian Optimisation scheme which both represents and optimises aleatoric noise in the suggestions. We consider cases such as drug discovery where we would like to minimise or be robust to aleatoric uncertainty but also applications such as materials discovery where it may be beneficial to maximise or be antifragile to aleatoric uncertainty. Our scheme features a heteroscedastic Gaussian Process (GP) as the surrogate model in conjunction with two acquisition heuristics. First, we extend the augmented expected improvement (AEI) heuristic to the heteroscedastic setting and second, we introduce a new acquisition function, aleatoric-penalised expected improvement (ANPEI) based on a simple scalarisation of the performance and noise objective. Both methods are capable of penalising or promoting aleatoric noise in the suggestions and yield improved performance relative to a naive implementation of homoscedastic Bayesian Optimisation on toy problems as well as a real-world optimisation problem.

## 1 INTRODUCTION

Bayesian Optimisation is already being utilised to make decisions in high-stakes applications such as drug discovery [1, 2, 3, 4, 5], materials discovery [6, 7, 8], robotics [9], sensor placement [10] and tissue engineering [11]. In these problems heteroscedastic or input-dependent noise is rarely accounted for and the assumption of homoscedastic noise is often inappropriate. As a case study, heteroscedastic noise is the rule rather than the exception in the majority of scientific datasets. This is the case, not only in experimental datasets, but also in datasets where properties are predicted computationally. We illustrate this for molecular hydration free energies in Figure 1 using the dataset of [12] where there is a distribution of noise values and in general the noise function might be expected to grow in proportion to chemical complexity [13]. The consequences of neglecting heteroscedastic noise are illustrated using a second example in Figure 2. The homoscedastic model will underestimate noise in certain regions which could induce a Bayesian Optimisation scheme to suggest values possessing large aleatoric noise. In an application such as high-throughput virtual screening the cost of misrepresenting the noise during the screening process could amount to a year wasted in the physical synthesis of a drug [14].

In materials discovery, we may derive benefit from or be antifragile [15] towards high aleatoric uncertainty. In an application such as the search for high-performing perovskite solar cells, we are faced with an extremely large compositional space, with millions of potential candidates possessing high aleatoric noise for identical reproductions[16]. In this instance we may want to guide search towards a candidate possessing a high photoluminescence quantum efficiency with high aleatoric noise. If the cost of repeating material syntheses is small relative to the cost of the search, the large aleatoric noise will present opportunities to synthesise materials possessing efficiencies far above their mean values.

In this paper we present a heteroscedastic Bayesian Optimisation scheme capable of both representing and optimising aleatoric noise in the suggestions. Our contributions are:

1. The introduction of a novel combination of surrogate model and acquisition function designed to optimise heteroscedastic aleatoric uncertainty to be used in situations where one wants to minimise (be robust to) aleatoric uncertainty as well as situations where one wants to maximise (be antifragile towards) aleatoric uncertainty.

2. A demonstration of our scheme's ability to outperform naive schemes based on homoscedastic Bayesian Optimisation on toy problems in addition to a real-world optimisation problem.

3. The provision of an open-source implementation of the model.

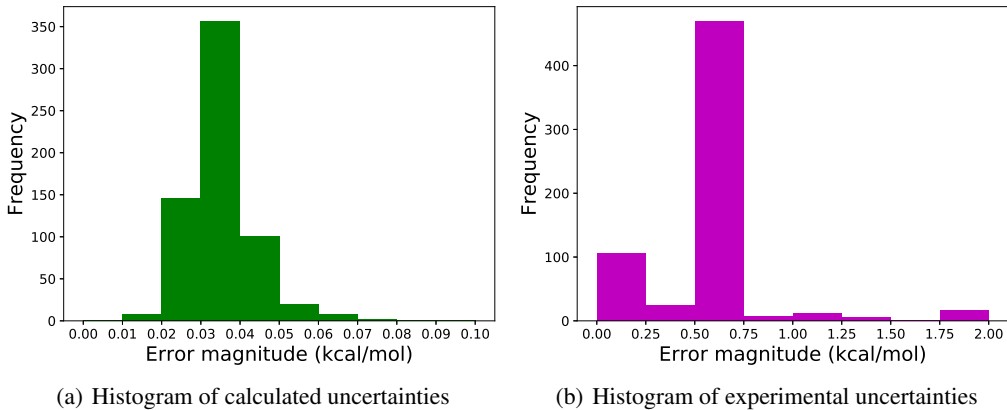

(a) Histogram of calculated uncertainties      (b) Histogram of experimental uncertainties

Figure 1: Fig. (a) shows the frequency histogram of the calculated uncertainties against the error magnitude (kcal/mol) for the FreeSolv hydration energy dataset ([12]). Fig. (b) shows a similar plot for the experimental uncertainties.

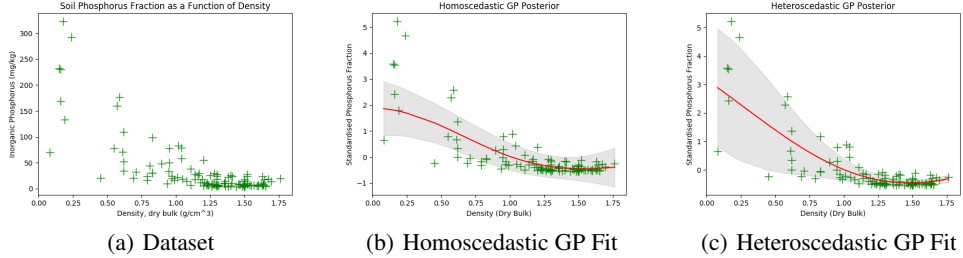

(a) Dataset      (b) Homoscedastic GP Fit      (c) Heteroscedastic GP Fit

Figure 2: Comparison of Homoscedastic and Heteroscedastic GP Fits to the Soil Phosphorus Fraction Dataset.

## 2 RELATED WORK

The most similar work to our own is that of [17] where experiments are reported on a heteroscedastic Branin-Hoo function using the variational heteroscedastic GP approach of [18] although to the best of our knowledge this work does not consider sequential evaluations. A modification to EI, expected risk improvement is introduced in [19] and is applied to problems in robotics where robustness to aleatoric noise is desirable. [20, 21] implement heteroscedastic Bayesian Optimisation but don't introduce an acquisition function that penalises aleatoric uncertainty. [22, 23] consider the related problem of safe Bayesian Optimisation through implementing constraints in parameter space. In this instance, the goal of the algorithm is to enforce a performance threshold for each evaluation of the black-box function and so is unrelated to our problem definition. In terms of acquisition functions,

[24, 25] propose principled approaches to handling aleatoric noise in the homoscedastic setting that could be extended to the heteroscedastic setting. Our primary focus in this work however, is to highlight that heteroscedasticity in the surrogate model is beneficial and so an examination of a subset of acquisition functions is sufficient for this purpose.

## 3 BACKGROUND

Bayesian Optimisation features a surrogate model for the black-box objective. The surrogate model should maintain calibrated uncertainty estimates in order to guide the acquisition of new data points. GPs are often a popular choice of surrogate model in BO applications because of their ability to represent uncertainty.

### 3.1 GAUSSIAN PROCESSES

In the terminology of stochastic processes we may formally define a GP as follows:

**Definition 1.** A Gaussian Process is a collection of random variables, any finite number of which have a joint Gaussian distribution.

The random variables consist of function values $f(\mathbf{x})$ at different locations $\mathbf{x}$ within the design space. The GP is characterised by a mean function

$$m(\mathbf{x}) = \mathbb{E}[f(\mathbf{x})] \tag{1}$$

and a covariance function

$$k(\mathbf{x}, \mathbf{x}') = \mathbb{E}[(f(\mathbf{x} - m(\mathbf{x}))(f(\mathbf{x}') - m(\mathbf{x}')))]. \tag{2}$$

The process is written as follows

$$f(\mathbf{x}) \sim \mathcal{GP}\big(m(\mathbf{x}), k(\mathbf{x}, \mathbf{x}')\big). \tag{3}$$

In our experiments, the prior mean function will be set to the data mean after normalisation of the inputs and standardisation of the outputs. As such, $m(\mathbf{x}) = 0$ will be assumed henceforth. The covariance function computes the pairwise covariance between two random variables (function values). The covariance between a pair of output values $f(\mathbf{x})$ and $f(\mathbf{x}')$ is a function of an input pair $\mathbf{x}$ and $\mathbf{x}'$. As such, the kernel encodes smoothness assumptions about the latent function being modelled; similarity in input space yields outputs that are close [26]. The most widely-utilised kernel is the squared exponential (SE) kernel

$$k_{\text{SQE}}(\boldsymbol{x}, \boldsymbol{x}') = \sigma_f^2 \cdot \exp\Big(\frac{-\|\boldsymbol{x} - \boldsymbol{x}'\|^2}{2\ell^2}\Big) \tag{4}$$

where $\sigma_f^2$ is the signal amplitude hyperparameter (vertical lengthscale) and $\ell$ is the (horizontal) lengthscale hyperparameter. We use the squared exponential kernel in all experiments. For further information on Gaussian Processes the reader is referred to [27].

### 3.2 BAYESIAN OPTIMISATION

**Problem Statement** The global optimisation problem is defined as

$$\mathbf{x}^* = \underset{\mathbf{x} \in \mathcal{X}}{\arg\min} \, f(\mathbf{x}) \tag{5}$$

where $\mathbf{x}^*$ is the global optimiser of a black-box function $f : \mathcal{X} \to \mathcal{Y}$. $\mathcal{X}$ is the design space and is typically a compact subset of $\mathbb{R}^d$. What makes this optimisation problem practically relevant in applications are the following properties:

1. Black-box Objective: We do not have the analytic form of $f$. We can however evaluate $f$ pointwise anywhere in the design space $\mathcal{X}$.

2. Expensive Evaluations: Choosing an input location $\mathbf{x}$ and evaluating $f(\mathbf{x})$ takes a very long time.

3. Noise: The evaluation of a given $\mathbf{x}$ is a noisy process. In addition, this noise may vary across $\mathcal{X}$, making the underlying process heteroscedastic.

We have a dataset of past observations $\mathcal{D} = \{(\boldsymbol{x}_i, t_i)\}_{i=1}^n$ consisting of observations of the black-box function $f$ and fit a surrogate model to these datapoints. We then leverage the predictive mean as well as the uncertainty estimates of the surrogate model to guide the acquisition of the next data point $\boldsymbol{x}_{n+1}$ according to a heuristic known as an acquisition function.

## 4    HETEROSCEDASTIC BAYESIAN OPTIMISATION

We wish to perform Bayesian Optimisation whilst optimising input-dependent aleatoric noise. In order to represent input-dependent aleatoric noise, a heteroscedastic surrogate model is required. We take the most likely heteroscedastic GP approach of [28], adopting the notation presented there for consistency. We have a dataset $\mathcal{D} = \{(\mathbf{x}_i, t_i)\}_{i=1}^n$ in which the target values $t_i$ have been generated according to $t_i = f(\mathbf{x}_i) + \epsilon_i$. We assume independent Gaussian noise terms $\epsilon_i \sim \mathcal{N}(0, \sigma_i)$ with variances given by $\sigma_i = r(\mathbf{x}_i)$. In the heteroscedastic setting $r$ is typically a non-constant function over the input domain $\mathbf{x}$. In order to perform Bayesian Optimisation, we wish to model the predictive distribution $P(\mathbf{t}^* \mid \mathbf{x}_1^*, \ldots, \mathbf{x}_q^*)$ at the query points $\mathbf{x}_1^*, \ldots, \mathbf{x}_q^*$. Placing a GP prior on $f$ and taking $r(\mathbf{x})$ as the assumed noise rate function, the predictive distribution is multivariate Gaussian $\mathcal{N}(\mu^*, \Sigma^*)$ with mean

$$\mu^* = E[\mathbf{t}^*] = K^*(K + R)^{-1}\mathbf{t} \tag{6}$$

and covariance matrix

$$\Sigma^* = \mathrm{var}[\mathbf{t}^*] = K^{**} + R^* - K^*(K + R)^{-1}K^{*T}, \tag{7}$$

where $K \in \mathbb{R}^{n \times n}$, $K_{ij} = k(\mathbf{x}_i, \mathbf{x}_j)$, $K^* \in \mathbb{R}^{q \times n}$, $K_{ij}^* = k(\mathbf{x}_i^*, \mathbf{x}_j)$, $K^{**} \in \mathbb{R}^{q \times q}$, $K_{ij}^{**} = k(\mathbf{x}_i^*, \mathbf{x}_j^*)$, $\mathbf{t} = (t_1, t_2, \ldots, t_n)^T$, $R = \mathrm{diag}(\mathbf{r})$ with $\mathbf{r} = (r(\mathbf{x}_1), r(\mathbf{x}_2), \ldots, r(\mathbf{x}_n))^T$, and $R^* = \mathrm{diag}(\mathbf{r}^*)$ with $\mathbf{r}^* = (r(\mathbf{x}_1^*), r(\mathbf{x}_2^*), \ldots, r(\mathbf{x}_q^*))^T$.

The most likely heteroscedastic GP algorithm [28] executes the following steps:

1. Estimate a homoscedastic GP, $G_1$ on the dataset $\mathcal{D} = \{(\mathbf{x}_i, t_i)\}_{i=1}^n$

2. Given $G_1$, we estimate the empirical noise levels for the training data using $\mathrm{var}[t_i, G_1(\mathbf{x}_i, \mathcal{D})] = 0.5\,(t_i - \mathbb{E}[x])^2$ forming a new dataset $\mathcal{D}' = \{(\mathbf{x}_i, z_i)\}_{i=1}^n$. A note on the form of this variance estimator is give in Appendix B.

3. Estimate a second GP, $G_2$ on $\mathcal{D}'$.

4. Estimate a combined GP, $G_3$ on $\mathcal{D}$ using $G_2$ to predict the logarithmic noise levels $r_i$.

5. If not converged, set $G_3$ to $G_1$ and repeat.

The Bayesian Optimisation problem may be framed as

$$\boldsymbol{x}^* = \arg\min_{\boldsymbol{x} \in \chi} f(\boldsymbol{x}), \tag{8}$$

where the black-box objective $f$, to be minimised has the form

$$f(\boldsymbol{x}) = g(\boldsymbol{x}) + s(\boldsymbol{x}). \tag{9}$$

$s(\boldsymbol{x})$ is, in this instance, the true noise rate function. We investigate extensions of the expected improvement [29] acquisition criterion, the principal form of which may be written in terms of the targets $t$ and the incumbent best objective function value, $\eta$, found so far as

$$\text{EI}(\boldsymbol{x}) = \mathbb{E}\big[\,(\eta - t)_+\big] = \int_{-\infty}^{\infty} (\eta - t)_+ \, p(t \,|\, \boldsymbol{x}) \, dt \tag{10}$$

where $p(t \,|\, \boldsymbol{x})$ is the posterior predictive marginal density of the objective function evaluated at $\boldsymbol{x}$. $(\eta - t)_+ \equiv \max(0, \, \eta - t)$ is the improvement over the incumbent best objective function value $\eta$. Evaluations of the objective are noisy in all of the problems we consider and so we use expected improvement with plug-in [30], the plug-in value being the GP predictive mean [31].

We propose two extensions to the expected improvement criterion. The first is an extension of the augmented expected improvement criterion

$$\text{AEI}(\boldsymbol{x}) = \mathbb{E}\big[(\eta - t)_+\big] \left( 1 - \frac{\sigma_n}{\sqrt{\text{var}[\mathbf{t}^*] + \sigma_n^2}} \right), \tag{11}$$

of [32] where $\sigma_n$ is the fixed aleatoric noise level. We extend AEI to the heteroscedastic setting by exchanging the fixed aleaotric noise level with the input-dependent one:

$$\text{het-AEI}(\boldsymbol{x}) = \mathbb{E}\big[(\eta - t)_+\big] \left( 1 - \frac{\sqrt{r(\boldsymbol{x})}}{\sqrt{\text{var}[\mathbf{t}^*] + r(\boldsymbol{x})}} \right). \tag{12}$$

where $r(\boldsymbol{x})$ is the predicted aleatoric uncertainty at input $\boldsymbol{x}$ under the most likely heteroscedastic GP and $\text{var}[\mathbf{t}^*]$ is the predictive variance of the heteroscedastic GP incorporating both aleatoric and epistemic components of the uncertainty. We also propose a simple modification to the expected improvement acquisition function that explicitly penalises regions of the input space with large aleatoric uncertainty. We call this acquisition function aleatoric noise-penalised expected improvement (ANPEI) and denote it

$$\text{ANPEI} = \alpha \text{EI}(\boldsymbol{x}) - (1 - \alpha)\sqrt{r(\boldsymbol{x})} \tag{13}$$

where $\alpha$ is a scalarisation constant which we set to $0.5$ for the experiments in this paper. In the multiobjective optimisation setting a particular value of $\alpha$ will correspond to a point on the Pareto frontier. We use both the modification to AEI (het-AEI) and ANPEI acquisition function in conjunction with the most likely heteroscedastic GP surrogate model in the experiments that follow.

For the experiments on antifragility to aleatoric uncertainty we propose the antifragile augmented expected improvement (AAEI) acquisition function

$$\text{AAEI}(\boldsymbol{x}) = \mathbb{E}\big[(\eta - t)_+\big] \left( 1 - \frac{\sqrt{\text{var}[\mathbf{t}^*] + r(\boldsymbol{x})}}{\sqrt{r(\boldsymbol{x})}} \right). \tag{14}$$

which has the effect of promoting solutions with high aleatoric noise.

## 5 Experiments on Robustness to Aleatoric Uncertainty

### 5.1 Implementation

Experiments were run using a custom implementation of Gaussian Process regression and most likely heteroscedastic Gaussian Process regression. The code is available at https://anonymous.

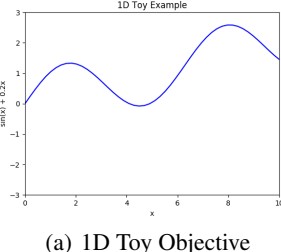 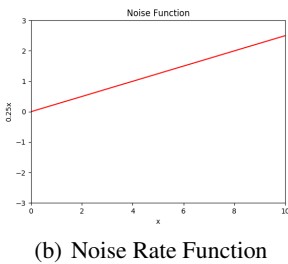 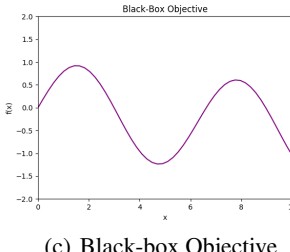

(a) 1D Toy Objective     (b) Noise Rate Function     (c) Black-box Objective

Figure 3: Toy 1D Problem. The toy objective in a) is corrupted with heteroscedastic noise according to the function in b). The combined objective, which when optimised maximises the sin wave subject to the minimisation of aleatoric noise, is given in c) and is obtained by subtracting the noise function from the 1D sinusoid.

4open.science/r/3361287c-6879-4b38-9153-5c9491271200/. The squared exponential kernel was chosen as the covariance function for both the homoscedastic GP as well as $G_1$ and $G_2$ of the most likely heteroscedastic GP. The lengthscales, $\ell$, of the homoscedastic GP were initialised to 1.0 for each input dimension across all toy problems after standardisation of the output values following the recommendation of [33]. The signal amplitude $\sigma_f^2$ was initialised to a value of 2.5. The lengthscale, $\ell$, of $G_2$ of the most likely heteroscedastic GP [28] was initialised to 1.0, the initial noise level of $G_2$ was set to 1.0. The EM-like procedure required to train the most likely heteroscedastic GP was run for 10 iterations and the sample size required to construct the variance estimator producing the auxiliary dataset was 100. Hyperparameter values were obtained by optimising the marginal likelihood using the scipy implementation of the L-BFGS optimiser. The objective function in all cases is the principal objective $g(\boldsymbol{x})$ minus one standard deviation of the ground truth noise function $s(\boldsymbol{x})$.

## 5.2 1D Toy Objective with Linear Noise Rate Function

Referring to Equation 9 from section 4, in the first experiment we take a one-dimensional sin wave

$$g(x) = \sin(x) + 0.2(x) \tag{15}$$

with noise rate function $s(x) = 0.25x$. These functions as well as the black-box objective $f(x)$ are shown in Figure 3. The Bayesian Optimisation problem is designed such that the first maximum in 3(a) is to be preferred as samples from this region of the input space will have a smaller noise rate. The black-box objective in 3(c) illustrates this trade-off. In 6(a) we compare the performance of a Bayesian Optimisation scheme involving a vanilla GP in conjunction with the EI acquisition function with the most likely heteroscedastic GP in conjunction with the ANPEI and het-AEI acquisition functions. The experiment is designed to contrast the performance of a standard Bayesian Optimisation scheme against our approach in a situation where minimising aleatoric noise is desirable.

## 5.3 Branin-Hoo with Non-linear Noise Rate Function

In the second experiment we consider the Branin-Hoo function as $g(\boldsymbol{x})$ with a non-linear noise rate function given by $s(\boldsymbol{x}) = 1.4x_1{}^2 + 0.3x_2$. Given that this example is a minimisation problem, the black-box objective consists of the sum of the Branin-Hoo function and the noise rate function. Contour plots of the functions are shown in Figure 4. A comparison, in terms of the best objective function values found, between the vanilla GP and EI acquisition function with the most likely heteroscedastic GP and ANPEI and AEI acquisition functions is given in 6(b).

## 5.4 Optimising the Phosphorus Fraction of Soil

In this real-world problem we apply heteroscedastic Bayesian optimisation to the problem of optimising the phosphorus fraction of soil. Soil phosphorus is an essential nutrient for plant growth and is

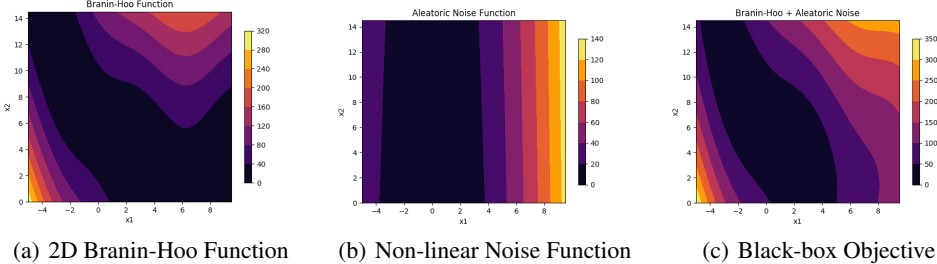

(a) 2D Branin-Hoo Function  (b) Non-linear Noise Function  (c) Black-box Objective

Figure 4: Toy 2D Problem. The Branin-Hoo objective function in a) is corrupted by the heteroscedastic noise function in b) $s(x_1, x_2) = 1.4{x_1}^2 + 0.3x_2$. The black-box objective function c) is obtained by summing the functions in a) and b). The sum is required to penalise regions of large aleatoric noise because the objective is being minimised.

widely used as a fertilizer in agriculture. While the amount of arable land worldwide is declining, global population is expanding and so is food demand. As such, understanding the availability of plant nutrients that increase crop yield is essential. To this end, [34] have curated a dataset on soil phosphorus, relating phosphorus content to variables including soil particle size, total nitrogen, organic carbon and bulk density. In this experiment, we study the relationship between bulk soil density and the phosphorus fraction, the goal being to minimise the phosphorus content of soil subject to heteroscedastic noise. We provide evidence that there is heteroscedasticity in the problem by comparing the fits of a homoscedastic GP and the most likely heteroscedastic GP in Figure 2 and provide a predictive performance comparison based on negative log predictive density values in the appendix. In this problem, we do not have access to a continuous-valued black-box function or a ground truth noise function. As such, the surrogate models were initialised with a subset of the data and the query locations selected by Bayesian Optimisation were mapped to the closest data points in the heldout data. The following kernel smoothing procedure was used to generate pseudo ground truth noise values:

1. Fit a homoscedastic GP to the full dataset.
2. At each point $x_i$, compute the corresponding square error $s_i^2 = (y_i - \mu(x_i))^2$.
3. Estimate variances by computing a moving average of the squared errors, where the relative weight of each $s_i^2$ was assigned with a Gaussian kernel.

The performances of homoscedastic Bayesian Optimisation using EI and AEI and heteroscedastic Bayesian Optimisation using ANPEI and AEI are compared in 5(c).

## 5.5 DISCUSSION

In all robustness experiments, the most likely heteroscedastic GP and ANPEI combination/heteroscedastic GP and het-AEI combination outperform the homoscedastic GP and EI. The fact that the homoscedastic GP has no knowledge of the heteroscedasticity of the noise rate function puts it at a serious disadvantage. In the first sin wave problem, designed to highlight this point, the heteroscedastic Bayesian Optimisation scheme consistently and preferentially finds the first maximum as that which minimises aleatoric noise. In contrast the homoscedastic GP, finds it impossible to differentiate between the two maxima. The experiments provide strong evidence that modelling heteroscedasticity in Bayesian Optimisation is a more flexible approach to assuming homoscedastic noise.

## 6 EXPERIMENTS ON ANTIFRAGILITY TO ALEATORIC UNCERTAINTY

### 6.1 IMPLEMENTATION

The implementational details remain the same as for the robustness experiments save for the fact that the acquisition functions have changed such as to promote antifragility to aleatoric uncertainty.

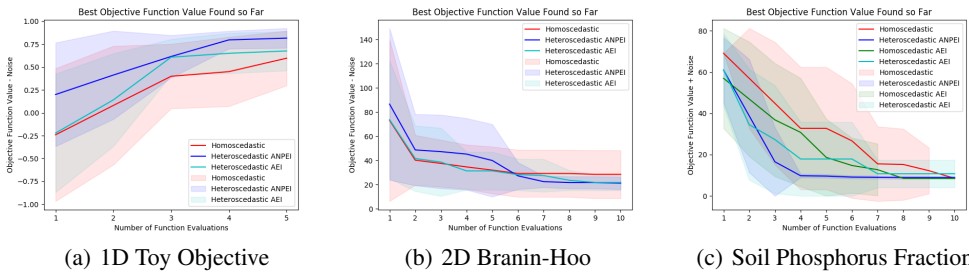

(a) 1D Toy Objective     (b) 2D Branin-Hoo     (c) Soil Phosphorus Fraction

Figure 5: Results of heteroscedastic and homoscedastic Bayesian Optimisation on the 3 robustness problems considered. Error bars are computed using 10 random initialisations. The first problem is a maximisation problem whereas the second and third are minimisation problems.

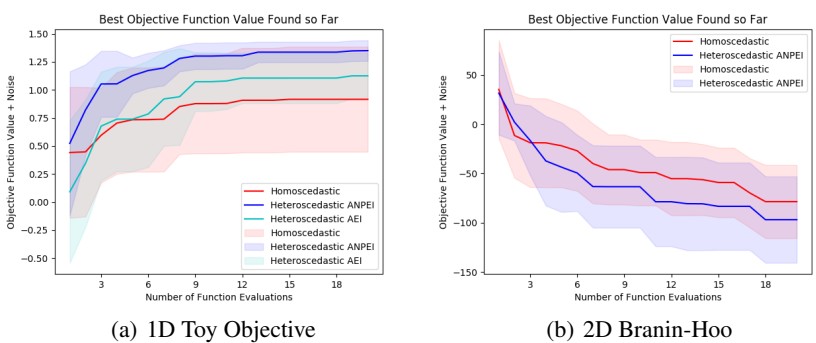

(a) 1D Toy Objective         (b) 2D Branin-Hoo

Figure 6: Results of heteroscedastic and homoscedastic Bayesian Optimisation on the sin and Branin toy problems where antifragility to aleatoric uncertainty is desirable. Error bars are computed using 10 random initialisations. The first problem is a maximisation problem whereas the second is a minimisation problem. Heteroscedastic AEI in this instance is the AAEI acquisition function.

This was achieved by changing the sign in the expression for ANPEI and using the AAEI acquisition function in place of het-AEI. The baseline homoscedastic GP and EI combination remains the same. The homoscedastic GP is incapable of representing input-dependent aleatoric uncertainty and so is unable to incorporate information about this quantity in any acquisition function that is used in conjunction with it.

## 6.2 DISCUSSION

The results of Figure 6 demonstrate that heteroscedastic Bayesian Optimisation is capable of seeking out solutions possessing high aleatoric noise, a desirable property in materials discovery applications. The scalarisation heuristic used in the modified ANPEI acquisition function outperforms the AAEI acquisition function on the toy sin function example suggesting that this approach may be both valid and beneficial in its simplicity in certain scenarios.

## 7 CONCLUSION AND FUTURE WORK

We have presented an approach for performing Bayesian Optimisation with the explicit goal of optimising aleatoric uncertainty in the suggestions. We posit that such an approach can prove useful for the natural sciences in the search for molecules and materials that are robust and antifragile to experimental noise and in future work we plan to apply our approach to molecular property optimisation [4]. We demonstrate concrete improvements on one and two-dimensional toy problems as well as a real-world optimisation problem and contribute an open-source implementation of the most likely heteroscedastic GP as a surrogate model for Bayesian Optimisation.

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

## A    HETEROSCEDASTICITY OF THE SOIL PHOSPHORUS FRACTION DATASET

Table 1 is used to demonstrate the efficacy of modelling the soil phosphorus fraction dataset using a heteroscedastic GP. The heteroscedastic GP outperforms the homoscedastic GP on prediction based on the metric of negative log predictive density (NLPD)

$$\text{NLPD} = \frac{1}{n} \sum_{i=1}^{n} -\log p(t_i | \boldsymbol{x_i}) \tag{16}$$

which penalises both over and under-confident predictions.

Table 1: Comparison of NLPD values on the soil phosphorus fraction dataset. Standard errors are reported for 10 independent train/test splits.

| Soil Phosphorus Fraction Dataset | GP | Het GP |
|---|---|---|
| NLPD | $1.35 \pm 1.33$ | $1.00 \pm 0.95$ |

