# OpenReview forum: "Antifragile and Robust Heteroscedastic Bayesian Optimisation"
_ICLR.cc/2020/Conference — Reject_

### Official Review · AnonReviewer3 · 2019-10-16
**Official Blind Review #3**

**Rating:** 3

**Review:**

The paper considers the heterogeneous noise in Bayesian optimisation. The paper utilised the existing heterogeneous Gaussian process to model the surrogate function and proposes two acquisition functions from the Expected improvement to deal with such heterogeneous noise.

The proposed acquisition function is heuristic and straightforward from the existing ones, ie. augmented EI and EI. I would not count this as much in terms of novelty.

The idea of dealing with heterogeneous noise in BO is interesting.

The related background in Gaussian process is standard and could be omitted.

The experimental section is weak. The experiments are demonstrated using low dimensional functions (1-2 dim).

The experiment needs to compare with Random baseline. Under the noise setting, the Random approach performs very competitive to BO.
The y-axis in the experiment, the paper has considered the objective function value + noise. The reviewer suspects that the high/low performance is due to the high level of noise (?) rather than the better objective function value.
The released source code includes the Lidar, Scallop, Silverman datasets, but not the PHOSPHORUS soil. I would encourage the authors to check the source code before releasing.
The paper focuses on demonstrating the heteroscedasticity in the surrogate model using [1]. The reviewer is wondering what is the performance for other heteroscedastic GP approaches, such as [2,3,4] ?

Minor points:
Page 4, step number 2, what is $z_i$ and how can we get it in a new dataset D’.


[1] Kersting, Kristian, et al. "Most likely heteroscedastic Gaussian process regression." Proceedings of the 24th international conference on Machine learning. ACM, 2007.
[2] Le, Quoc V., Alex J. Smola, and Stéphane Canu. "Heteroscedastic Gaussian process regression." Proceedings of the 22nd international conference on Machine learning. ACM, 2005.
[3] Binois, M., Gramacy, R. B., & Ludkovski, M. (2018). Practical heteroscedastic gaussian process modeling for large simulation experiments. Journal of Computational and Graphical Statistics, 27(4), 808-821.
[4] Lázaro-Gredilla, M., & Titsias, M. K. (2011, June). Variational Heteroscedastic Gaussian Process Regression. In ICML (pp. 841-848).


**Experience Assessment:**

I have published in this field for several years.

**Review Assessment: Checking Correctness Of Derivations And Theory:**

I carefully checked the derivations and theory.

**Review Assessment: Checking Correctness Of Experiments:**

I assessed the sensibility of the experiments.

**Review Assessment: Thoroughness In Paper Reading:**

I read the paper at least twice and used my best judgement in assessing the paper.

---

### Official Review · AnonReviewer2 · 2019-10-23
**Official Blind Review #2**

**Rating:** 1

**Review:**

The main contribution of this paper are:

1. The use of a heteroscedastic GP when performing Bayesian Optimization, this is in contrast to the more common practice of assuming homoscedastic noise, even when this does not quite fit the data. They use the existing algorithm called most likely heteroscedastic GP, and quote previous work that performed BO using different heteroscedastic GP implementations.
2. They introduce two new acquisition functions that incorporate the predicted observation noise, either making candidates more likely or less likely to be chosen when predicted noise is higher, depending on the requirements. This are fairly minor extensions/heuristics if taken on their own, as they do not provide a very strong motivation indicating why these acquisition functions are useful or better than existing ones, other than that they take heteroscedasticity into account.
3. They run a set of experiments on the above settings. Unfortunately the experiments are very limited, and their method does not improve on the baselines in a statistically significant way. Two of the experiments are on simple synthetic settings, the third approximates a real world setting, although the approximation is quite rough and they don't convincingly argue for it being realistic, neither do they give convincing motivation of their objective which uses g +- standard deviation.
4. They provide source code of their implementation.

The paper is easy to understand, and covers an interesting topic, so while I don't think it meets the bar of ICLR (due to lack of convincing and non-trivial contributions) I think it could perhaps be made into a workshop submission with some of the following changes:
* A wider set of experimental settings, and more replication such that any differences become statistically significant. It would also be worthwhile comparing to random search.
* A better justification of the objective used in the experiments, using g +- the standard deviation appears fairly arbitrary, and there is no strong enough reason to believe this is a good approximation of what the cost is in the case of real world problems.
* Better theoretical justification of the acquisition function; one option is to introduce more principled acquisition function like, say, expected upper/lower bound.


Other notes/comments:
abstract: as well as a real-world -> as well as *on* a real-world...
section 1 "As a case study" -> not very clear what this means
incumbent best is not well defined, is it the empirical value of f used or the mean predicted f on the evaluated candidates?
section 6.2 "outperforms" -> this is not clear if one looks at the confidence intervals in the results


**Experience Assessment:**

I have read many papers in this area.

**Review Assessment: Checking Correctness Of Derivations And Theory:**

I assessed the sensibility of the derivations and theory.

**Review Assessment: Checking Correctness Of Experiments:**

I assessed the sensibility of the experiments.

**Review Assessment: Thoroughness In Paper Reading:**

I read the paper at least twice and used my best judgement in assessing the paper.

---

### Official Review · AnonReviewer1 · 2019-10-23
**Official Blind Review #1**

**Rating:** 1

**Review:**

SUMMARY OF REVIEW

This paper discusses an interesting problem of BO in the cases of robustness and antifragility to aleatoric noise/uncertainty. To tackle this problem, the authors replace the conventional homoscedastic GP model with a heteroscedastic GP model. In the case of robustness to aleatoric noise/uncertainty, the authors have modified EI by simply either scaling down [32] or subtracting from its value more when the aleatoric noise increases. In the case of antifragility to aleatoric noise/uncertainty, they do the opposite.

The modifications of EI to handle robustness and antifragility to aleatoric noise/uncertainty are simple and straightforward, one of which is similar to the augmented EI of [32].

There are some technical ambiguities, as detailed below. In particular, the choice of objective function (equation 9) for this problem needs to be justified and motivated by the practical applications described in Section 1.

Since no convergence guarantee is given, a more extensive empirical analysis with real datasets needs to be provided to better understand the performance and behavior of the proposed BO algorithms. In particular, though the authors have motivated their problem using the compelling applications of materials and drug discovery, no experimental result for these applications has been provided to support the motivation of this work.



DETAILED COMMENTS

The authors seem to motivate the significance of their problem of interest through the key applications of materials and drug discovery which I can appreciate. Unfortunately, experimental results in such applications were not available in this paper to "close the loop" in supporting the motivation of this work, begging the question whether their proposed BO algorithm indeed works for these key applications. For example, why is the FreeSolv hydration energy dataset not used for your experiments?


For Fig. 1, how exactly do you extract the error magnitudes from the FreeSolv hydration energy dataset? How do you exactly define the notion of calculated vs. experimental uncertainties? Fig. 1 shows that the noise peaks with a relatively high frequency at a single error magnitude value. Would the assumption of homoscedastic noise at this peaked value be detrimental to BO? A sensitivity analysis would be useful here.


For the soil phosphorus fraction dataset (Fig. 2), the skewed distribution of the measurements (few extremely large measurements and many small-valued measurements) may not be due to heteroscedastic aleatoric uncertainty. In fact, in the literature of earth/environmental science, such a dataset is often modeled using a log-Gaussian process (or log-normal kriging), that is, the log-measurements follow a GP:

Webster, R., and Oliver, M. 2007. Geostatistics for Environmental Scientists. John Wiley & Sons, 2nd edition.

Can the authors provide supporting evidence (in the form of references) that such a dataset is due to heteroscedastic aleatoric uncertainty?



On page 4, step 2 of the most likely heteroscedastic GP algorithm [28] cannot be understood: What is E[x]? Isn't G_1 a GP? Why is it able to accept x_i and D as inputs? How is z_i defined?

The authors say that "A note on the form of this variance estimator is give in Appendix B." There is no Appendix B.


Can the authors give a detailed discussion why is the expression of f(x) = g(x) + s(x) in equation 9 the right one to be minimized in practice (e.g., in the context of materials and drug discovery)? For example, do material scientists use such an objective function? Provide references. Furthermore, why is the same equation 9 being minimized for both the cases of robustness and antifragility to aleatoric uncertainty?

Caption of Fig. 3: I can't understand the sentence: "The combined objective, which when optimised maximises the sin wave subject to the minimisation of aleatoric noise". This was repeated in Section 5.5: "finds the first maximum as that which minimises aleatoric noise". Isn't the minimum of the aleatoric noise at the origin (Fig. 3b)?

Can the authors provide information on how much initial data was provided prior to running BO? How much data is used for learning the GP hyperparameters?

The performance advantage of heteroscedastic ANPEI over homoscedastic does not appear to be significant for Branin-Hoo function (Figs. 5b and 6b). Can the authors explain this?



Minor issues

There are two different font types of x in bold.

Page 5: fixed aleaotric

Which phrase is correct? "is obtained by subtracting the noise function from the 1D sinusoid" in the caption of Fig. 3 or "The objective function in all cases is the principal objective g(x) minus one standard deviation of the ground truth noise function s(x)"?

Fig. 5b: Shouldn't the vertical axis be labeled as ...+ Noise?

Fig. 6a: Shouldn't the vertical axis be labeled as ...- Noise?

**Experience Assessment:**

I have published in this field for several years.

**Review Assessment: Checking Correctness Of Derivations And Theory:**

I carefully checked the derivations and theory.

**Review Assessment: Checking Correctness Of Experiments:**

I carefully checked the experiments.

**Review Assessment: Thoroughness In Paper Reading:**

I read the paper thoroughly.

---

### Author Response · Authors · 2019-11-15
**Author Response to Reviewers**

We thank the reviewers for their helpful comments and feedback. We will endeavour to incorporate the suggestions into a more comprehensive and extended work to be submitted to another venue.

---

### Decision · Program_Chairs · 2019-12-19

**Decision:**

Reject

**Comment:**

The reviewers initially gave scores of 1,1,3 citing primarily weak empirical results and a lack of theoretical justification.  The experiments are presented on synthetic examples, which is a great start but the reviewers found that this doesn't give strong enough evidence that the methods developed in the paper would work well in practice.  The authors did not submit an author response to the reviewers and as such the scores did not change during discussion.  This paper would be significantly strengthened with the addition of experiments on actual problems e.g. related to drug discovery which is the motivation in the paper.